# Breast Collagen Organization: Variance by Patient Age and Breast Quadrant

**DOI:** 10.3390/diagnostics14161748

**Published:** 2024-08-12

**Authors:** Arnold Caleb Asiimwe, Monica Pernia Marin, Mary Salvatore

**Affiliations:** Department of Radiology, Columbia University Irving Medical Center, 177 Fort Washington Avenue, New York, NY 10032, USA; aa4870@columbia.edu (A.C.A.); monicapernia.md@gmail.com (M.P.M.)

**Keywords:** breast density, breast cancer, chest CT

## Abstract

Breast density is an important marker for increased breast cancer risk, but the ideal marker would be more specific. Breast compactness, which reflects the focal density of collagen fibers, parallels breast cancer occurrence being highest in the upper outer quadrants of the breast. In addition, it peaks during the same time frame as breast cancer in women. Improved biomarkers for breast cancer risk could pave the way for patient-specific preventive strategies.

## 1. Introduction

The incidence of breast cancer is increasing. Approximately 300,000 new cases of breast cancer will be diagnosed this year [1]. There are nearly 4 million current breast cancer survivors in the United States [1]. There is a one in eight lifetime risk for a woman to develop breast cancer [1]. Efforts to diagnose breast cancer early with mammography have only minimally reduced the rate of advanced breast cancer occurrence [2]. The emphasis needs to shift towards preventing breast cancer. The highest known relative risk for breast cancer is associated with an age greater than 65 [1]. Mammographically dense breast parenchyma has a relative risk of 2.1 to 4.0, which is the same as the relative risk associated with having Ductal carcinoma in situ or two or more first-degree relatives with breast cancer and so breast density is an important potentially modifiable risk factor [1]. The challenge remains that many women with dense breast parenchyma will not develop breast cancer, some women with minimal breast parenchyma will develop breast cancer.

Chemoprevention includes treatments to lower a woman’s risk of developing breast cancer, and includes receptor modulators such as tamoxifen and aromatase inhibitors, which may decrease breast density [3,4]. A tool to assess breast cancer risk could guide who would benefit most from chemopreventive treatment, but it needs to be more specific than breast density. We investigated breast compactness on chest CT, which is a measure of focal breast collagen organization, to see if it paralleled the most frequent age of breast cancer occurrence and the most common location of breast cancer, which is in the upper outer aspect of the breasts. Breast density is the extent of fibroglandular tissue in the breast. Breast compactness is the organization of a specific area of breast tissue and reflects the degree of collagen. Increased collagen deposition causes increased breast density. The degree of collagen organization in breast tissue is reflected by the measurement of Hounsfield Units (HU), a measure of density, on chest computed tomography (CT). More tightly organized collagen with higher HU is stiffer, and stiffness is pro-carcinogenic [5].

## 2. Materials and Methods

This HIPAA-compliant, IRB-approved retrospective study utilized a catalyst to identify women 30 years of age and older who had a non-contrast chest CT scan. The woman’s age was established based on the date of the CT scan. Exclusion criteria included prior breast surgery and non-visualization of the complete breast on imaging. The list of patients was sorted by age. Ten sequential women in each decade who had a non-contrast chest CT were included in this study. The breast was divided into quadrants following nipple localization. Upper outer and upper inner were above the nipple line, and lower outer and inner were below the level of the nipple. In each area, the brightest section was identified, and a manual tool, provided by the CT manufacturer, of roughly the same area was used to measure the maximum HU value of that quadrant.

## 3. Results

Maximum HUs per quadrant were documented (Figure 1 and Figure 2). The average max HU for all patients in all quadrants was 44. The average maximum HU for the right and left upper outer quadrants were highest at 62 and 58 HU, respectively. The average maximum HU for the right and left lower inner quadrants was the lowest, at 29 HU (Figure 3 and Figure 4). Maximum HU over 65 occurred in 55% of upper outer quadrant measurements and only 6% of lower inner quadrant measurements. A total of 90% of women in their 7th decade had measurements of greater than 65 HU in their outer quadrant, in contrast to 40% in their 3rd and 9th decades, when breast cancer is less common (Table 1). Very high HU numbers of 90 and above mostly occur in the 4th to eighth decade and in the upper outer aspects of the breast. Maximum HU numbers less than 60 were more common in the inner aspect of the breast, where breast cancer is less frequent (Table 2).

## 4. Discussion

Breast density is defined as the extent of fibro-glandular tissue that appears white on mammography [6]. Patients with high mammographic density are at greater risk of developing breast cancer than those with low mammographic density [7,8,9]. Not all women with dense breast parenchyma on mammography develop breast cancer, and not all women without dense breast parenchyma are spared. In general, a woman’s breast density decreases as she ages, yet her risk for breast cancer increases, thus generating conflicting statements [6]. Breast compactness, a new concept, is the measure of the maximum HUs of the breast on a chest CT. Maximum HUs reflect radiodensity, and in areas with the highest HUs, the breast collagen is most compact. Compact collagen causes breast stiffness, ultimately leading to a change in cell proliferation and gene expression with the potential to create and support an invasive cancer phenotype [5]. Women with more breast parenchyma are statistically more likely to have areas that have very high maximum Hus, which might explain why mammographic breast density is associated with increased breast cancer risk. Breast compactness has the opportunity to be more precise at predicting risk because even women with low breast density can have areas of compact breast tissue.

While breast density decreases as a woman ages, breast compactness does not. Breast density is related to the extent of breast stroma, or collagen, over the entire breast. High and low breast density tissue have the same quantity of epithelial elements, but the stromal elements are increased with high breast density [10,11]. Aging causes the collagen fibers to become not only thinner but also curvier, causing a denser mesh that is stiffer. Increased stiffness promotes breast cancer [12]. Collagen density can cause breast cancer initiation and progression [13,14,15]. Extracellular matrix, which is aged, can transform breast epithelial cells to a cancer-like phenotype [16].

Breasts become less dense with age; however, twisting of the collagen increases stiffness, which can be measured with elastography [17]. Chest CT provides another opportunity to indirectly measure stiffness by manual assessment of HUs that reflect the stromal density. Higher HUs are associated with more organized collagen and increased lysyl oxidase [18,19,20]. Thus, we expected that older women would have higher maximum HUs that correspond to the peak ages for breast cancer occurrence (50–70 years old) [1].

In our review of each decade, there were women with very high breast compactness (≥90 HU), and the rise and fall of the number of women with the most compact breast tissue paralleled the increased and decreased incidence of breast cancer reported in the literature. That is to say that few women under 30 have very compact breast tissue despite their high breast density. The women with the greatest incidence of areas of compact breast tissue are middle-aged, which mirrors breast cancer risk, and the number decreases above the age of 80.

Breast cancer is more common in the upper outer quadrant (51%), compared to the upper inner quadrant (14%) [21]. Therefore, we expected that the highest HUs would be in the upper outer breast tissue. Our review of women per decade shows that the highest percentage of high HUs are located in the outer quadrants of the breast, similar to breast cancer location.

The strength of collagen is affected by lysyl oxidase (LOX) crosslinks [22]. LOX enhances the stiffness of the extracellular matrix. Aging upregulates the expression of LOX [23]. Treatments targeting LOX can decrease matrix stiffness and potentially decrease breast cancer risk and aggressiveness [24]. Further research may reveal if measurements of maximum HUs in patients with breast cancer correlate with tumor aggressiveness. The tumor-to-stroma ratio (TSR) is the ratio of tumor cells to stromal cells on light microscopy; tumors with high stroma have poor outcomes, and stroma increases with age [25].

Currently, breast density is considered one of the strongest predictors of breast cancer risk. Breast compactness could be even more specific and identify not only who is at risk but where cancer is most likely to occur. This would encourage the further development of preventive breast cancer strategies, and their success could be monitored by follow-up of the breast compactness after intervention. This could theoretically lead to a significant advance in the fight against breast cancer mortality. Prospective studies are needed to evaluate the utility of screening for breast cancer with a chest CT. Retrospective studies have shown promising results [26]. Potentially, breast compactness on chest CT could identify those most at risk for breast cancer, screen for existing tumors, evaluate axilla for adenopathy, and comment on tumor potential for aggressiveness.

Limitations of this study include the relative paucity of patients studied. Future work may use automated methods for quantifying the highest HU that will be more accurate and reproducible. Perhaps we should have also measured the retro-areolar area separately due to its unique association with breast cancer occurrence.

In conclusion, breast compactness as measured by maximum HU on non-contrast chest CT reflects local collagen organization, which could be a potential biomarker for women at the highest risk for breast cancer. Our research has demonstrated that it corresponds to both age groups and quadrants for breast cancer occurrence. Further research will be necessary to determine if breast compactness assists in risk stratification and if it is a modifiable risk factor.

## Figures and Tables

**Figure 1 diagnostics-14-01748-f001:**
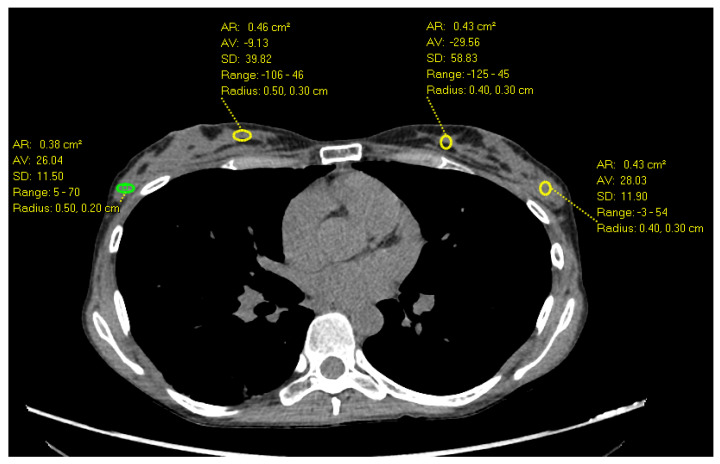
Axial Images of a non-contrast chest CT in a 32-year-old female with heterogeneously dense breast parenchyma demonstrates the highest HU measurements in the outer breast (70 HU) compared to inner breast.

**Figure 2 diagnostics-14-01748-f002:**
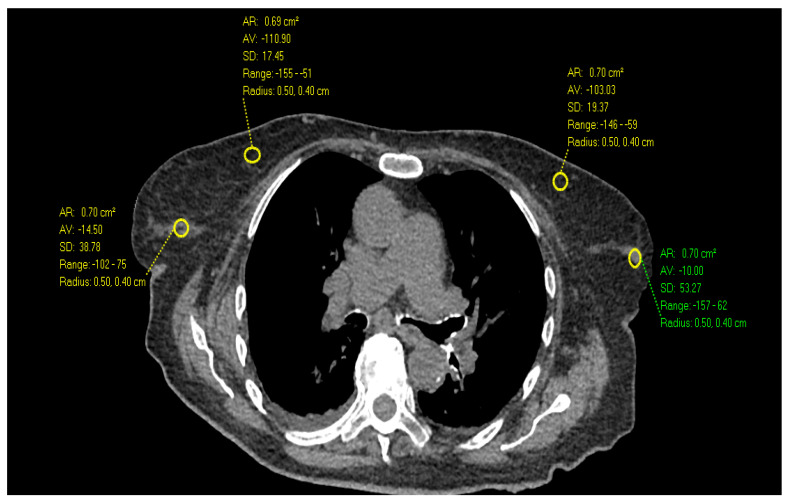
Axial Images of a non-contrast chest CT in a 90-year-old female with scattered fibroglandular tissue demonstrates the highest HU measurements in the outer breast (75 HU) compared to inner breast and higher than the 32-year-old female in Figure 1.

**Figure 3 diagnostics-14-01748-f003:**
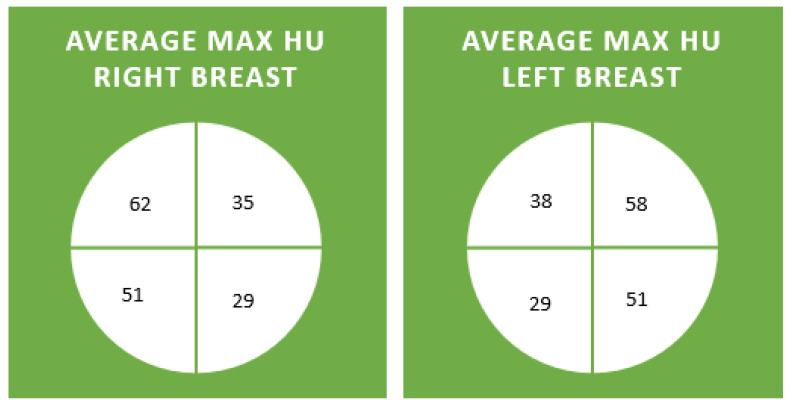
The average maximum HU in each breast for each quadrant across decades.

**Figure 4 diagnostics-14-01748-f004:**
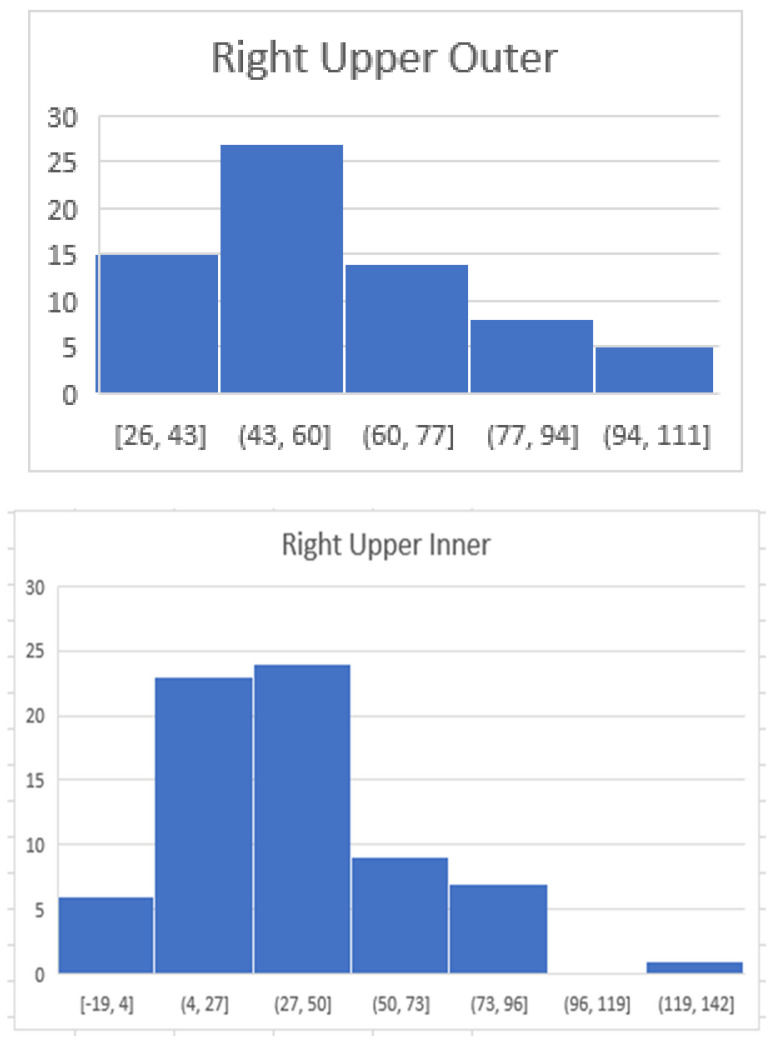
The following diagrams demonstrate the distribution of maximum HU per quadrant. The majority of maximum HU measurements in the right upper outer quadrant were from 43 to 60 HU whereas in the right lower inner quadrant they were 16–35. The x axis reflects the HU and the Y axis reflects the number of patients within the specified range.

**Table 1 diagnostics-14-01748-t001:** Percentage of women with High HU (≥65) by quadrant and decade.

	Outer Upper	Outer Lower	Inner Upper	Inner Lower
3rd decade	40%	20%	20%	20%
4th decade	50%	30%	20%	0%
5th decade	40%	30%	30%	0%
6th decade	45%	45%	15%	10%
7th decade	90%	10%	0%	0%
8th decade	80%	20%	0%	0%
9th decade	40%	40%	10%	10%
Total average	55%	27%	14%	6%

**Table 2 diagnostics-14-01748-t002:** Overall data of maximum HU with very High HU (≥90) highlighted in yellow and lower HU (<60) highlighted in green.

	3rd Decade	4th Decade	5th Decade	6th Decade	7th Decade	8th Decade	9th Decade
RUO Max	54 (35–74)	68 (35–111)	62 (26–99)	59 (29–97)	71 (36–204)	58 (31–101)	63 (36–106)
LUO Max	44 (13–71)	62 (44–79)	58 (16–86)	47 (27–84)	78 (19–195)	57 (36–85)	52 (41–63)
RLO Max	39 (21–64)	57 (21–92)	57 (34–94)	60 (29–112)	45 (22–76)	47 (18–104)	53 (43–60)
LLO Max	40 (−17–62)	62 (18–108)	51 (4–78)	40 (23–86)	46 (7–72)	44 (4–97)	59 (35–73)
RUI Max	40 (9–69)	43 (1–91)	33 (1–123)	38 (−19–86)	30 (-5–59)	30 (−1–79)	34 (7–95)
LUI Max	45 (26–81)	42 (18–70)	34 (4–80)	41 (11–84)	32 (1–84)	34 (12–80)	35 (21–58)
RLI Max	33 (5–47)	36 (−15–81)	21 (−10–58)	27 (−14–71)	18 (−22–46)	32 (9–86)	33 (12–60)
LLI Max	24 (−10–101)	46 (3–75)	29 (4–55)	32 (8–81)	10 (−33–46)	22 (−9–76)	31 (4–63)

## Data Availability

The data presented in this study are available on request from the corresponding author. The data are not publicly available due to retrospective nature and lack of informed consent.

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
