# Peer review of "Breast Collagen Organization: Variance by Patient Age and Breast Quadrant"

_diagnostics, 2024, doi:10.3390/diagnostics14161748_

Round 1

Reviewer 1 Report

Comments and Suggestions for Authors

Dear Authors, 

Your topic is very interesting and it could represent a new way to approach breast cancer detection and prevention. Nevertheless, I have some concerns.

Most of your references are old and some of them (e.g. reference n.2) has been cited to justify new approaches to breast cancer screening and prevention, giving incorrect information to the readers. Consequently, I think that it would be appropriate to cite more recent sources and to better justify the reason to evaluate breast compactness. The first part of the introduction (breast cancer epidemiology) should be synthesized and the whole section should be focused more on the basis to evaluate the suggested concept of breast compactness. Indeed, also the discussion section should elaborate also on the possible impact of this evaluation on clinical practice. You suggested that further evaluations are needed to correlate breast cancer compactness to tumor aggressiveness but I think that, if you consider this aspect as a risk factor, you should evaluate how to use it in clinical practice. Is possible a comparison whit BI-RADS system About this I have some concerns because chest CT has higher dose levels in comparison to mammography which is the golden standard for early diagnosis. Moreover, why have you used CT and not MRI? Is possible a comparison whit BI-RADS system? 

Materials and methods are not described clearly. How many women have you evaluated? What does it mean "ten sequential women"? What is the "manual tool" used for the measurement of HU? Please, provide more details and separate the description of the tables from the content of the section. It should be rewritten more clearly.  

This research has been funded or supported in any way? Please disclose of justify potential conflict of interests.

Author Response

Reviewer #1

Your topic is very interesting and it could represent a new way to approach breast cancer detection and prevention.

Thank you sincerely for your kindness and valuable insights. I have learned from each of them and worked to improve the value of the paper.

Nevertheless, I have some concerns. Most of your references are old and some of them (e.g. reference n.2) has been cited to justify new approaches to breast cancer screening and prevention, giving incorrect information to the readers. Consequently, I think that it would be appropriate to cite more recent sources and to better justify the reason to evaluate breast compactness.

I have updated the manuscript to include these 6 more recent manuscripts.

  • Hussain A, Bourguet-Kondracki ML, Hussain F, Rauf A, Ibrahim M, Khalid M, Hussain H, Hussain J, Ali I, Khalil AA, Alhumaydhi FA, Khan M, Hussain R, Rengasamy KRR. The potential role of dietary plant ingredients against mammary cancer: a comprehensive review. Crit Rev Food Sci Nutr. 2022;62(10):2580-2605. 
  • Flugelman AA, Burton A, Keinan-Boker L, Stein N, Kutner D, Shemesh L, Boyd N. Correlation between cumulative mammographic density and age-specific incidence of breast cancer: A biethnic study in Israel. Int J Cancer. 2022 Jun 15;150(12):1968-1977.
  • Nazari SS, Mukherjee P. An overview of mammographic density and its association with breast cancer. Breast Cancer. 2018 May;25(3):259-267. 
  • Li X, Jin Y, Xue J. Unveiling Collagen's Role in Breast Cancer: Insights into Expression Patterns, Functions and Clinical Implications. Int J Gen Med. 2024 May 2;17:1773-1787. 
  • Northey JJ, Hayward MK, Yui Y, Stashko C, Kai F, Mouw JK, Thakar D, Lakins JN, Ironside AJ, Samson S, Mukhtar RA, Hwang ES, Weaver VM. Mechanosensitive hormone signaling promotes mammary progenitor expansion and breast cancer risk. Cell Stem Cell. 2024 Jan 4;31(1):106-126.e13.
  • Liburkin-Dan T, Toledano S, Neufeld G. Lysyl Oxidase Family Enzymes and Their Role in Tumor Progression. Int J Mol Sci. 2022 Jun 2;23(11):6249.

The first part of the introduction (breast cancer epidemiology) should be synthesized and the whole section should be focused more on the basis to evaluate the suggested concept of breast compactness.

We have added the section below expanding on the concept of breast compactness.

Increased collagen deposition causes increased breast density. The degree of collagen organization in breast tissue is reflected by the measurement of Hounsfield Units (HU), a measure of density, on chest CT. More tightly organized collagen with higher HU are stiffer and stiffness is pro-carcinogenic.

Indeed, also the discussion section should elaborate also on the possible impact of this evaluation on clinical practice.

Currently, breast density is considered one of the strongest predictors of breast cancer risk. Breast compactness could be even more specific and identify not only who is at risk but where cancer is most likely to occur. This would encourage the further development of preventive breast cancer strategies and their success could be monitored by follow-up of the breast compactness after intervention. This could theoretically lead to a significant advance in the fight against breast cancer mortality.

You suggested that further evaluations are needed to correlate breast cancer compactness to tumor aggressiveness but I think that, if you consider this aspect as a risk factor, you should evaluate how to use it in clinical practice.

Prospective studies are needed to evaluate the utility of screening for breast cancer with chest CT. Retrospective studies have shown promising results (Desperito E, Schwartz L, Capaccione KM, Collins BT, Jamabawalikar S, Peng B, Patrizio R, Salvatore MM. Chest CT for Breast Cancer Diagnosis. Life (Basel). 2022 Oct 26;12(11):1699.). Potentially breast compactness on chest CT could identify those most at risk for breast cancer, screen for existing tumor, evaluate axilla for adenopathy and comment on tumor potential for aggressiveness.

Is possible a comparison whit BI-RADS system About this I have some concerns because chest CT has higher dose levels in comparison to mammography which is the golden standard for early diagnosis.

Our hospitals mammograms are performed on dedicated mammography units (Senographe Essential, GE Healthcare, Milwaukee, WI, USA). The views obtained consisted of the standard mediolateral oblique and craniocaudal views. For image acquisition, the parameters were as follows: kVp 29, mAs 35–77, and the mean effective dose of 0.72 mSv for all four views. The mean effective doses for low dose CTs at our site are 0.51 mSv.

Moreover, why have you used CT and not MRI? Is possible a comparison whit BI-RADS system? 

MRI is not able to measure HU.

Materials and methods are not described clearly. How many women have you evaluated? What does it mean "ten sequential women"? What is the "manual tool" used for the measurement of HU? Please, provide more details and separate the description of the tables from the content of the section. It should be rewritten more clearly.  

This HIPAA-compliant, IRB-approved retrospective study utilized catalyst to identify women 30 years of age and older who had a non-contrast CT scan. The woman's age was established based on the date of the CT scan. Exclusion criteria included prior breast surgery and non-visualization of the complete breast on imaging. The list of patients was sorted by age. 10 sequential women in each decade who had a non-contrast chest CT were included in this study. The breast was divided into quadrants following nipple localization. Upper outer and upper inner were above the nipple line and lower outer and inner were below the level of the nipple. In each area the brightest section was identified and a manual tool, provided by CT manufacturer, of roughly the same area was used to measure the maximum HU value of that quadrant.

This research has been funded or supported in any way? Please disclose of justify potential conflict of interests.

This research has not been funded. The results of this research will be used to apply for future funding to study breast compactness on chest CT and its role in risk stratification.

Reviewer 2 Report

Comments and Suggestions for Authors

This retrospective study is quite relevant and appropriate. The investigators used a non-contrast chest CT to determine whether increased breast compactness was associated with the most common location of breast cancer.  The study is compromised by the manual approach used to quantify the HU values, which was the only tool used. Lack of fixing the ROI area size impacts the mean, max, min and SD of the HU values within an ROI. Unless this fixed, I don’t see any of findings meaningful nor reliable. Enclosed below are some general and specific comments for the authors to consider.

General comments

1. There was no line numbering in the received PDF, which makes hard and less effective in pinpointing comments.    

2. From a biology and physiology stand point, what the difference between breast density and breast compactness?

3. One year between a screening mammogram and non-contrast CT chest might not reflect the same. One year gap is sufficient to develop any pathogeneses. What kind of measures you take to eliminate the impact on your findings?

Specific comments:

1.Title: patient decade, refers to what exactly? Patient age!! Please rephrase the title to reflect on the study itself.

2. Introduction: the second sentence in the introduction should be supported with a reference.  

3. Introduction: DCIS, MRI and CT are acronyms that need to be spelled out in the first appearance.

4. Materials and methods: a non-contrast CT. Do you mean chest CT, if yes then be specific.

5. Material sand methods: rephrase this sentence so it reads clear “Ten sequential women in each decade who had a non-contrast chest CT …..”. Do you mean women in their 30s, 40s, 50, etc.? Also, what is the 10 sequential referring to?

6. Material sand methods: authors need to elaborate more on how the HU measurement were taken starting with identifying the region of interest (ROI) area size, how to make sure the ROI is placed on the same location in the axial image etc. Changes in the ROI area and placement affects the mean, max, min and SD of HU. This is clear in Fig 1! Some of the HU difference between left and right are due to the ROI area size.

7. Fig 1: SD in one of the ROI drawn in the left breast was high because the region included is heterogeneous which also affects the mean, max and min HU.

8. Fig 4: Y and X axis should be labeled.

Author Response

Reviewer #2

This retrospective study is quite relevant and appropriate. The investigators used a non-contrast chest CT to determine whether increased breast compactness was associated with the most common location of breast cancer.  

Thank you sincerely for your valuable comments and the opportunity to improve the value of the paper for your readers.

The study is compromised by the manual approach used to quantify the HU values, which was the only tool used. Lack of fixing the ROI area size impacts the mean, max, min and SD of the HU values within an ROI. Unless this fixed, I don’t see any of findings meaningful nor reliable. Enclosed below are some general and specific comments for the authors to consider.

Thank you for pointing out this important observation. The only value of importance is the max HU and the region of interest was drawn with the goal of obtaining the highest HU and that was recorded. The area does not affect this measurement only the location of its placement. I have addended the writing to reflect these changes….

This HIPAA-compliant, IRB-approved retrospective study utilized catalyst to identify women 30 years of age and older who had a non-contrast CT scan. The woman's age was established based on the date of the CT scan. Exclusion criteria included prior breast surgery and non-visualization of the complete breast on imaging. The list of patients was sorted by age. 10 sequential women in each decade who had a non-contrast chest CT were included in this study. The breast was divided into quadrants following nipple localization. Upper outer and upper inner were above the nipple line and lower outer and inner were below the level of the nipple. In each area the brightest section was identified and a manual tool, provided by CT manufacturer, of roughly the same area was used to measure the maximum HU value of that quadrant.

General comments

There was no line numbering in the received PDF, which makes hard and less effective in pinpointing comments.    

I do not have the ability to change this and it is likely specific for the journal.

  1. From a biology and physiology stand point, what the difference between breast density and breast compactness?

Breast density is the extent of fibroglandular tissue in the breast. Breast compactness is the organization of a specific area of breast tissue and reflects the degree of collagen.

  1. One year between a screening mammogram and non-contrast CT chest might not reflect the same. One-year gap is sufficient to develop any pathogeneses. What kind of measures you take to eliminate the impact on your findings?

 We had only looked at the chest CT and so the statement regarding the mammogram was unnecessary and we have removed it.

Specific comments:

1.Title: patient decade, refers to what exactly? Patient age!! Please rephrase the title to reflect on the study itself.

Thank you for this valuable change…

Breast Collagen Organization: Variance by Patient Age and Breast Quadrant

  1. Introduction: the second sentence in the introduction should be supported with a reference.  

Thank you, the first 3 sentences are information from reference number 1. I have added a superscript.

The incidence of breast cancer is increasing. Approximately 300,000 new cases of breast cancer will be diagnosed this year1. There are nearly 4 million current breast cancer survivors in the United States1. There is a 1 in 8 lifetime risk for a woman to develop breast cancer1.

  1. Introduction: DCIS, MRI and CT are acronyms that need to be spelled out in the first appearance.

Thank you, we have updated manuscript to reflect the recommended change.

  1. Materials and methods: a non-contrast CT. Do you mean chest CT, if yes then be specific.

Thank you, we studied non-contrast chest CT scans and have updated manuscript to reflect that.

  1. Materials and methods: rephrase this sentence so it reads clear “Ten sequential women in each decade who had a non-contrast chest CT …..”. Do you mean women in their 30s, 40s, 50, etc.? Also, what is the 10 sequential referring to?

This HIPAA-compliant, IRB-approved retrospective study utilized catalyst to identify women 30 years of age and older who had a non-contrast CT scan. The woman's age was established based on the date of the CT scan. Exclusion criteria included prior breast surgery and non-visualization of the complete breast on imaging. The list of patients was sorted by age. 10 sequential women in each decade who had a non-contrast chest CT were included in this study.

  1. Materials and methods: authors need to elaborate more on how the HU measurement were taken starting with identifying the region of interest (ROI) area size, how to make sure the ROI is placed on the same location in the axial image etc. Changes in the ROI area and placement affects the mean, max, min and SD of HU. This is clear in Fig 1! Some of the HU difference between left and right are due to the ROI area size.

The breast was divided into quadrants following nipple localization. Upper outer and upper inner were above the nipple line and lower outer and inner were below the level of the nipple. In each area the brightest section was identified and a manual tool, provided by CT manufacturer, of roughly the same area was used to measure the maximum HU value of that quadrant.

  1. Fig 1: SD in one of the ROI drawn in the left breast was high because the region included is heterogeneous which also affects the mean, max and min HU.

We only consider the maximum HU and so it is not affected by the variation.

  1. Fig 4: Y and X axis should be labeled

The x axis reflects the HU and the Y axis reflects the number of patients within the specified range. I have added this to the figure description.

Round 2

Reviewer 1 Report

Comments and Suggestions for Authors

Comments addressed.

Reviewer 2 Report

Comments and Suggestions for Authors

Most of the comments were satisfactory addressed by the authors.